# Insights into the Microstructural Evolution Occurring during Pyrolysis of Metal-Modified Ceramers Studied through Selective SiO_2_ Removal

**DOI:** 10.3390/ma14123276

**Published:** 2021-06-14

**Authors:** Aitana Tamayo, Juan Rubio, Fausto Rubio, Mᵃ Angeles Rodriguez

**Affiliations:** 1Institute of Ceramics and Glass, CSIC, Kelsen 5, 28049 Madrid, Spain; jrubio@icv.csic.es (J.R.); frubio@icv.csic.es (F.R.); 2Faculty of Science, Department of Environmental, University of Extremadura, Avda. Elvas s/n, 06071 Badajoz, Spain; marodgon@unex.es

**Keywords:** polymer-derived ceramics, ceramer, molecular precursors, HF etching, pyrolysis, spectroscopy

## Abstract

Silicon oxycarbide ceramers containing 5% aluminum, zirconium, and cobalt with respect to the total Si amount are prepared from a commercial polysiloxane and molecular precursors and pyrolyzed at temperatures ranging from 500 to 1000 °C. HF etching is carried out to partially digest the silica phase, thus revealing structural characteristics of the materials, which depend upon the incorporated heteroatom. From the structural and textural characterization, it was deduced that when Al enters into the ceramer structure, the crosslinking degree is increased, leading to lower carbon domain size and carbon incorporation as well. On the contrary, the substitution by Zr induced a phase-separated SiO_2_-ZrO_2_ network with some degree of mesoporosity even at high pyrolysis temperatures. Co, however, forms small carbidic crystallites, which strongly modifies the carbonaceous phase in such a way that even when it is added in a small amount and in combination with other heteroatoms, this transient metal dominates the structural characteristics of the ceramer material. This systematic study of the ceramer compounds allows the identification of the ultimate properties of the polymer-derived ceramic composites.

## 1. Introduction

The partial conversion of polysiloxane precursors into hybrid materials (ceramers) with high specific surface areas, tunable pore size distributions, and adjustable surface characteristics has been widely studied over the past years. The chemical and physical properties can be easily modified by using different heteroatoms, leading to the production of completely new materials with high purity and homogeneous distribution of the constituent phases. Bonding the heteroatoms to the hybrid precursor before crosslinking as well as the use of coordination compounds are the most explored methodologies for the preparation of metal-modified polymer-derived ceramics and ceramers [1]. At the early stages of research in polymers and hybrid materials for nanostructured ceramics, much effort was made on the modification of a silicon-based gel by aluminum and boron [2,3,4]. Since then, and despite the high potential of these materials, the majority of the studies are based on incorporating various heteroatoms with the aim of increasing the operational temperature [5]. Nanocomposites, based on silicon polymers with metallic nanoparticles, are an exciting solution, enabling the development of sensors, chemical reactors, membrane supports, or thermal insulators, among many other applications [6,7,8,9].

The different phase compositions of the SiMOC ceramers (M = Ti, Al, Zr, Hf, Fe, Mn, etc.) has been demonstrated to be thermodynamically controlled by oxidation-reduction reactions, which depend on the type of heteroatom introduced into the ceramic network [10]. It is known that SiAlOC materials minimize the extent of the carbothermal reduction, compared to pure SiOC and can be employed as high-temperature pressure sensors [11,12]. The minimized amount of free carbon phase and SiO_4_ units, together with the formation of mullite, contribute to the reduction of the extent of the carbothermal reaction, thus permitting the development of novel high-temperature barrier resistant coatings [13]. The low activation energy of mullite formation through the polymer-derived ceramic route allows fast and quantitative crystallization of mullite at lower temperatures than the conventional methods. Moreover, its crystallization extent can be further controlled with the increase in temperature. Then, when mullite units appear, the carbon is segregated from the SiOC network via vapor–liquid–solid mechanism, producing a highly densified network, a reaction that is enhanced with the increase in the number of SiC units [14]. This extraordinary microstructure has been used for the development of piezoelectric ceramics in which the piezo-dielectricity of the material resulted from the formation of space charges along the interface between the relatively insulating ceramic phase and the more conductive free-carbon phase [15]. It must be highlighted that in piezoelectric materials, the presence of relatively conducting grains and insulating grain boundaries is considered to be responsible for the polarization mechanism; nevertheless, in the polymer-derived ceramics, the presence of these space charges is not yet completely understood and must be evaluated.

On the other hand, one of the first commercial applications of polymer-derived ceramics was the production of SiC carbon fibers; the SiTiCO fibers were the first-generation fibers commercialized by Ube Industries [16]. In order to reduce the oxygen content of the fibers, Ti was replaced by Zr, leading to the consecution of improved strength retention at high temperatures. Contrary to Ti, Zr does not usually form a TiC phase but ZrO_2_ instead, which afterward evolves to the ZrC phase, inducing enhanced thermal stability [17]. Now, the third generation of SiC fibers includes Al to help the crosslinking of the polymeric fibers. 

Transition-metal-based nanoparticles (Ni, Fe, and Co) deposited over the surface of a highly porous SiOC and SiCN matrix have also been investigated as efficient metal-supported catalysts for CO_2_ methanation because of the selective modification of the microstructure of the obtained materials [18,19]. The Ni-modified, polymer-derived ceramers result in a dispersion of small nanoparticles, well distributed throughout the network, where C tends to form large crystallites up to 10 nm when the ceramer materials are pyrolyzed at temperatures higher than 600 °C, resulting in a highly hydrophilic surface with 1D nanostructures [20]. The formation of these 1D carbon nanostructures within the pores of the ceramics has also been reported by several authors [9,21], showing an enhanced catalytic behavior. Co catalyst was more effective in the formation of the carbon nanostructures, increasing considerably the available specific surface area for reaction [22]. The growth mechanism depends on the included heteroatom [20,22,23,24]. 

The characterization of a silicon-based material after HF etching and subsequent oxidation allowed an increased number of the SiO_4_ tetrahedral to be identified, whereas the relative number of SiC_4_ tetrahedral diminished, indicating that the surface of the HF materials was completely heterogeneous [25]. After etching, remainders of Si-C on the surface of the graphitic carbon layers, which evolves to SiO_2_, turned out to be dependent on the pyrolysis temperature, thus highlighting the importance of studying the tendency to segregate SiO_2_ clusters during the thermal transformation [26]. The work described here aims to provide a comprehensive overview of the changes occurring during the pyrolysis of several metal-modified silicon-based ceramers, focusing on the carbon phase and the formation of the mixed SiOC network. 

The selection of the heteroatoms is based on the effect these atoms exert on the microstructure. While it is known that Al easily incorporates into the SiOC network, forming a SiC-enriched network, Co tends to form 1D and 2D C nanostructures. In addition, Zr is known to increase the thermal stability of the polymer-derived ceramics and the formation of ZrO_2_ nanoclusters, thus minimizing the number of Si atoms bonded to O. Here, the evolution of the microstructure is studied in a ceramer network deficient of Si-O bonds because of the addition of the heteroatom Zr. The matrix without Co is studied as well. Through chemical etching in a post-pyrolysis process [7,9], we have removed the SiO_2_-like clusters, allowing our materials to show the distinct features of the surface of the porous materials and the free carbon phase, as well as allowing the study of the main structural characteristics of the substituted networks [27,28]. Therefore, the final aim of the study is to provide a comprehensive overview of the changes induced in the ceramer material because of the presence of the different heteroatoms—alone and in combination—with a focus on identifying the ultimate reason for their intriguing characteristics.

## 2. Materials and Methods

Commercially available polymethylsilsesquioxane (Silres^®^ MK powder, Wacker Chemie, Munich, Germany, empirical formula (RSiO_1,5_)_n_ where R = CH_3_ and *n* = 130–150) was used as raw material. Zirconium acetylacetonate (Zr (Acac), Sigma-Aldrich, Burlington MA, USA), aluminum acetylacetonate (Al (Acac), Sigma-Aldrich, Burlington MA, USA), and cobalt acetate (CoAc, Probus, Esparraguera, Spain) were mixed with MK preceramic polymer in a planetary ball mill at 150 rpm for 15 min. The amount of Zr (Acac), Al (Acac), and CoAc was fixed in such a way that the heteroatom/Si weight ratio was 5%. ((Zr/Si) = (Al/Si) = (Co/Si) = 0.05). Once mixed, the materials were pyrolyzed in an alumina tube furnace at a fixed heating/cooling rate of 5 °C/min to temperatures between 500 and 1000 °C under a continuous nitrogen flow of 150 mL/min, 5 h dwelling at 280 °C, plus 2 h at the maximum temperature. Pyrolyzed samples were ground in an agate mortar and sieved using the 100–200 micrometer fraction for all analyses and treatments. In this work, sample labeling indicates the heteroatom used (or the two of them in the case of using two modifiers together, such as Zr and Co), followed by an F in the case of the etched samples.

Etching of the sieved samples was carried out by magnetic stirring in an HF solution (40% v/v) for 24 h, and the resulting materials were recovered by centrifugation at 2300 rpm for 12 min. Etching in powdered materials was preferred rather than in bulk samples to facilitate etching. No estimation of the etching efficiency was carried out. Then, the materials were washed with distilled water until the final pH of the washing liquors was 5.0. Etched powders were subsequently dried at 110 °C until constant weight.

The chemical composition was calculated from the chemical analysis carried out in elemental analyzers (LECO Corp, St Joseph MI, USA), CS-200 for the C determination, RC-420 for hydrogen content, and TC-500 to determine the oxygen amount. At least four analyses were carried out per sample and equipment, and the resulting value was calculated by average. Silicon was determined by difference. Structural characterization of the SiOC materials was carried out by means of FTIR and Raman spectroscopies. FTIR analyses were performed in a PerkinElmer (Waltham MA, USA) model BX spectrophotometer by diluting the samples in KBr and obtaining the spectra as the average of at least 32 collections with a spectral resolution of 2 cm^−1^. Raman characterization was carried out in a Renishaw InVia spectrophotometer by using the 514 nm Ar^+^ ion laser as an excitation source in the confocal mode. The spectra were the accumulated signal of 10 collections with an accumulation time of 10 s.

Differential scanning calorimetry analyses were performed in a piece of Q600 equipment (TA Instruments, New Castel, DE, USA), using about 20 mg of the sample under 100 mL/min airflow. The instrument was calibrated with Aluminum and Gold for temperatures and sapphire for Heat Flow calculation. 

Textural analyses were carried out in the Tristar equipment (Micromeritics, Norcross GA, USA). From the adsorption–desorption N_2_ isotherm, the porous properties and the specific surface area of all samples were determined by using the BJH and BET methods, respectively [29]. Prior to the analysis, all the samples were degassed at 120 °C for 18 h.

## 3. Results

The percentage amounts of carbon, oxygen, and hydrogen in each sample were determined to obtain the overall chemical formulae of the studied materials (Table 1). In these formulae, it is estimated that the weight of the heteroatom, calculated from the initial molar ratio Si/M (M = Al, Zr, Co), is maintained constantly during the pyrolysis. 

Figure 1a shows the trend of the ratio O/C, along with the pyrolysis temperature in all the materials. It is clear that this ratio varies with the elements incorporated into the ceramic network, and it should be different when using different ligands attached to the heteroatom. In contrast to the case of Al, in which a maximum in the O/C ratio is reached at the intermediate pyrolysis temperatures (700–900 °C), when the heteroatom is a transition metal, a gradual increase is observed in the O/C ratio with the increase in the pyrolysis temperature. This ratio is also slightly higher when the Co (either alone or in combination with Zr) is incorporated into the preceramic network leading to a major O content when both the Zr and Co are introduced with respect to the Al-containing materials. Similarly, in Figure 1b, the composition of the materials pyrolyzed at 1000 °C is represented as a ternary diagram, in which the H amount has not been taken into account. There, the previous observation is observed that the incorporation of Zr and Co (and both elements together) induces an increase in the amount of O incorporated into the ceramic network at the maximum pyrolysis temperature.

### 3.1. Structural Characterization of the Ceramer Materials

In Figure 2a, the FTIR spectra of all the samples pyrolyzed at 700 °C are plotted since at this temperature, the bands are more clearly detected. In the low-frequency region, the (O–Si–O) centered at 470 cm^−1^ is shifted to lower wavenumbers in the sample containing Co, indicating a highly tensioned structure [30]. The tensioned structure contains distorted O–Si–O bond angles with respect to the normal values found in the SiO_2_. Since the ceramic network is partially formed, apart from the vibration of the Si–O bonds at 790 cm^−1^, the incorporation of the mixed C–Si–O units induced the appearance of a new band centered at 810–845 cm^−1^, which shifts to high or low wavenumbers depending on the pyrolysis temperature and the incorporated heteroatom. Figure 2b presents the position of this band calculated by means of a deconvolution analysis. Here, it is shown that when Al or Zr is added, the SiOC band shifts to 845–835 cm^−1^, at pyrolysis temperatures comprised between 700 and 900 °C, and then decreases again to 810 cm^−1^, which is the reported position for SiOC glasses [31,32]. If the incorporated heteroatom is Co or Zr/Co, a gradual shift to high wavenumbers is produced when the pyrolysis is carried out up to 800 °C and then drops to 800–810 cm^−1^, which is considered the normal range for SiO_2_ and SiOC materials. Since the same trend is found in the two cases, it is reasonable to attribute the observed behavior to the presence of the Co. At 1000 °C, in the sample containing the two heteroatoms, the position of this band decreases to its lowest value, possibly due to phase separation.

The position and intensities of the IR bands vary with both the pyrolysis temperature and the incorporation of the heteroatom. A systematic study of the infrared spectra is carried out by performing the deconvolution analysis of the bands appearing in the spectra of all the studied samples (spectra shown in Appendix A). In the most intense region of the spectra, it is possible to distinguish the multiple combinations of the Si–O rings and Si–O–X [33,34], where X can be substituted by the heteroatom symbol. In this region, the band attributed to the stretching of Si–O bonds presents the highest relative intensity in the materials containing Al, followed by the materials containing Zr. This band is composed of the contribution of crosslinked SiO_4_ tetrahedral (tridimensional) and SiO_2_-like tetrahedral in form of SiO_2_ rings and chains. In the spectra, the band located at 1030 cm^−1^ is attributed to the ring-like superstructural units (TO_L/C_ transversal optic), and the tridimensional (TO_T_ transversal optic) SiO_2_ structures appeared at about 1080 cm^−1^ [35,36,37]. The plot of the relative intensity of the tridimensional SiO_2_ structures (TOT_,_ 1080 cm^−1^) to ring-like superstructural units (TO_L/C_, 1030 cm^−1^) is shown in Figure 2c, and it is observed that the relative intensity of this ratio undergoes low variations when either Al or Zr is incorporated into the hybrid network, but it dramatically increases when Co (and Co plus Zr) atoms are added at pyrolysis temperatures below 800 °C and then decreases. It should also be noticed that in the case of the Al incorporation, this band increases progressively up to 700 °C and afterward decreases. 

When the ceramic structure is not yet formed (low pyrolysis temperature), it is possible to appreciate the asymmetric stretching of vinyl groups at 1410 cm^−1^ [38,39] (Figure 2d). This band is the most noticeable in the samples containing Co because of the formation of highly graphitized carbon structures at 700–800 °C [40,41]. Beyond 900 °C, none of the samples except the one containing Zr present this characteristic feature.

For a better understanding of the structural characteristics of the materials and their evolution toward the ceramic state during the pyrolysis, we subject the samples to a chemical etching to remove the major part of the SiO_2_ phase. The Gaussian deconvolution of the FTIR spectra of the etched samples. Figure 3a is used to estimate the stability of the different ring-like or tridimensional SiO_2_ structures against the etching process. As we proceed with the un-etched samples, here (Figure 3b) we only show the spectra of the bands pyrolyzed at 900 °C because the contributions to the different bands are more evident. Despite that, it should also be noted here that we have performed the analysis on the samples pyrolyzed and HF etched at temperatures beyond 800 °C. By comparing Figure 2c and Figure 3b, the relative intensity of the TO_T_ to TO_L/C_ bands drastically decreases in all samples and at all the treatment temperatures, indicating that the tridimensional SiO_2_ is more susceptible to be etched away. The most dramatic change in the TO_T_/TO_L/C_ ratio is found in the samples containing Co in which the relative proportion of tridimensional units in the HF-etched samples decreases with the temperature. In the remainder materials, there is no relationship between the pyrolysis temperature and the amount of linear and tridimensional SiO_2_ before and after HF etching ((TO_T_/TO_L/C_)_pyrolyzed_/(TO_T_/TO_L/C_)_HF etch_). 

The effect of the HF etching is also observed in the carbon phase, as deduced from Raman spectroscopy (spectra shown in Appendix A). In Figure 4a, the characteristic Raman spectra of the HF-etched materials pyrolyzed at 900 °C is shown, with the D band appearing at 1350 cm^−1^, which is commonly attributed to the A_1g_ mode of the small graphite crystallites, and the G band, which is related with the in-plane bond stretching of sp^2^ bonds (E_2g_ mode) in carbon clusters [42]. Similar spectra are obtained for all the remaining samples. Here, it is clearly observed that the D and G bands become narrower in the materials containing Zr/Co and Co, indicating that the remainder carbon phase after the etching process is the more graphitized. In the spectra of the as-pyrolyzed samples, the I_D_/I_G_ ratio (Figure 4b) varies with both the temperature and the included heteroatom. For the calculations, and for considering the intensity of the band (I), we use the height of the Lorentzian-shaped band obtained from the deconvolution analysis, and background subtraction is carried out before the band deconvolution. In the case of the incorporation of Al and Zr, the I_D_/I_G_ ratio decreases with the temperature (i.e., the carbon phase becomes more “ordered”), whereas the incorporation of Co and Zr/Co provokes the opposite effect, with a decrease of the ordering of the carbonaceous phase. After etching, however, there is a slight increase in the I_D_/I_G_ ratio with the temperature when the heteroatom incorporated is Al. Since the D band only accounts for sp^2^ rings, the observed increase suggests that, during etching, both the C atoms in sp^2^ and sp^3^ configurations were etched away. 

### 3.2. Thermal Analysis

Valuable information about the carbon phase can be obtained when studying the thermal stability in the air, both before and after the etching treatment. The different behavior shown in all the samples is quite significant, with different decomposition temperatures and the number of decomposition stages depending on the substituent heteroatom (Figure 5a). The most characteristic temperatures during the oxidation of the materials are collected in Table 2 (differential thermal analysis curves are found in Appendix A). There, it should be noticed that despite the inorganic material is not formed (the materials are still in ceramer state), we are carrying out the decomposition in an oxidant atmosphere, i.e., preferentially the carbon phase. Solely in the materials containing Al and pyrolyzed at the different temperatures, the thermal decomposition occurs in a single step, at temperatures increasing from 450 to 650 °C, as the pyrolysis temperature increases. The samples containing Zr as the single heteroatom decompose in a single step when the materials are heat treated at temperatures beyond 700 °C, but two decomposition stages are found when they are pyrolyzed at 600 °C. By substituting the heteroatom for Co, now we can observe that the number of decomposition stages as well as the temperatures at which these reactions take place, depending upon the pyrolysis temperature. In Table 2, however, the most prominent peaks are exclusively collected.

The calculation of the oxidation enthalpy of the materials from the DSC curves (Figure 5b and Appendix A) allows the detection of the slight decrease in the decomposition enthalpy at pyrolysis temperatures below 800 °C, and then a severe decrease occurs at 900 °C. Oxidation enthalpy is calculated from the area under the DSC curve. This behavior is observed in the samples containing Al and Co but not in the materials containing Zr in which the oxidation enthalpy does not show significant variations among them.

### 3.3. Textural Characterization

It is well known that in most polymer-derived ceramics, transient porosity created in the intermediate stages of the pyrolysis (i.e., in the ceramer state) and associated with the elimination of gaseous species evolving from the decomposition of the preceramic matrix (CH_4_, H_2_, C_6_H_6_, etc.) [43]. This transient porosity is observed in all the synthesized materials. Table 3 summarizes the specific surface area of the pyrolyzed materials from the application of the BET method to the nitrogen adsorption isotherms. In all the materials, except in the case of the samples containing Zr as the unique heteroatom, this transient porosity disappears at 900 °C. 

The t-plot method [29] application to the N_2_ adsorption isotherms allows the calculation of the external surface (or area corresponding to the mesopores) from the slope of the t-plot (t-plots are provided in Appendix A). In Figure 6a, it is observed that the maximum external surface corresponds to the samples pyrolyzed at 600 °C and containing Al and Co, whereas, in the presence of Zr, there is a delay in the appearance of this porosity, which is attributed to enhanced preceramic network stability. With regard to the micropore surface area (Figure 6b), apart from the abovementioned porosity still remaining at 1000 °C in the material synthesized with Zr, it can be observed that the amount of micropores gradually increases, reaching the maximum at 700 °C. In the case of Co-containing samples, the increase of the microporosity to its maximum value occurs in a more drastic manner, and the temperature at which this maximum microporosity appears is different whether the sample is synthesized in the presence or the absence of Zr.

Table 3 also includes the specific surface area of some of the HF-etched materials. It is observed that the material presenting the highest SSA at 900 °C is the one synthesized with the heteroatom Al, with this SSA being even higher than in the corresponding pyrolyzed sample. The same occurs in the Co-containing material pyrolyzed at 900 °C, where the SSA corresponding to the HF-etched sample is higher than its non-etched counterpart. The pore size distributions calculated from the application of the BJH method to the desorption branch of the isotherms obtained in the HF-etched materials are provided in Appendix A. This behavior is attributed to the elimination of the stabilization of the SiO_2_ clusters, which favors phase separation and therefore the elimination of this component in the HF process. In the case of the Zr-containing materials, a pronounced decrease in the SSA is observed after the digestion process.

## 4. Discussion

Generally, the description of the structural changes induced by the heteroatoms within the hybrid network assumes a random and homogeneous distribution of the elements. The change in the pore volume occurring after the HF etching and attributed to the removal of some disordered C and SiOC units has already been reported [45]. The discontinuous distribution of the different phases because of the presence of the different heteroatoms is thus responsible for the observed variations in the ratio of the SiO_2_-like structure after the HF etching as well as the graphitization of the carbon phase.

It is known that during pyrolysis, Al is able to form the AlO_x_C_y_ at a relatively low temperature through the reaction of the Al(OH)_3_ particles, which are dispersed within the ceramer network with the free carbon, thus decreasing the amount of free carbon and the observed decrease in the O/C ratio at temperatures beyond 800 °C [2]. Moreover, the condensation mechanism of the Al-containing samples occurs via dehydrogenation reactions [46,47] with the formation of the C=C bonds at 700 °C (Figure 2d) [48]. At this temperature, the relative intensity of the TO_T_/TO_LC_ band is the highest as well. Zhang et al. [49] report the formation of Si-O-Al bridges during the pyrolysis at the expense of the Si-H bonds. At this temperature, and accompanied with the decrease of the Si-H band in the FTIR spectra (2100 cm^−1^), the position of the Si-O-C band is shifted from 817 cm^−1^ to its maximum value, at 845 cm^−1^, which is assigned to the symmetric stretching vibration of the Si–O–Si and Si-O-Al linkages [50]. The enhanced crosslinking at lower temperature with respect to the non-modified polysiloxane has already been described by many other authors, resulting in an increased proportion of highly crosslinked SiO_2_ (tridimensional) units [51,52]. This resulted in an increased intensity of the band centered at about 1100 cm^−1^, which contains the different modes of the Si–O–Si bonds (Figure 2a). In the HF-etched samples, the TO_T_ to TO_L/C_ ratio decreases with respect to their corresponding non-etched counterparts indicating that the SiO_2_ units, which were preferentially etched, are the ones in interconnected rings or chains. This result can be further corroborated from the SSA obtained after HF etching (Table 3). At 900 °C, there might be a large number of interconnected rings or chains that are preferably removed during the etching, leading to the appearance of large voids and thus increasing the SSA.

The decrease in the graphite nanodomain size of the pyrolyzed samples is attributed to the rearrangement of distorted aromatic carbon rings to six-membered rings [53]. However, after etching, the behavior of the I_D_/I_G_ ratio is exactly the opposite, suggesting that etching preferably affects the sp^3^ carbon. The differential thermal analysis data (Figure 5) show that the decomposition of the carbon phase occurs in just one single step, indicating that there are no secondary carbon phases evolving from this material and, as expected, the temperature at which this decomposition takes place gradually increases as the treatment temperature does. The decrease in the free-carbon phase content because of the cationic substitution of Si by Al atoms might be also responsible for the minimum oxidation enthalpy [48]. The presence of one type of carbon and the substitution of Si by Al atoms might cause the space charges and thus the polarization of the structure upon determined pressure and, ultimately, its piezoelectric behavior. This piezo-dielectricity of the material results from the formation of space charges along the interface between the relatively insulating ceramic phase and the more conductive free-carbon phase [15].

SiOC structural units are formed at temperatures as low as 700 °C either when the substituent heteroatom is Al or Zr; however, when the Zr is the heteroatom, some authors have reported that the relative amount of SiC units decreases with the Zr content [54]. Contrary to what occurs in Al- or Ti-containing SiOC glasses, during pyrolysis, it has been reported that amorphous ZrO_2_ clusters are segregated from the matrix [54]. In all the remaining samples, the TO_T_/TO_L/C_ ratio reaches a maximum of 700 °C, but the opposite trend is found in the Zr-modified SiOC. The phase separation occurring in these materials would be responsible for the decrease in Si-O-C formation and therefore, in the spectra of the HF samples (Figure 3a); we particularly highlight the presence of a single peak in the central band of the spectra because of the non-etched SiO_2_ units. 

During the synthesis of SiOC/Zr materials, the added Zr may act as an inert or active filler whenever the ZrO_2_ nanoclusters are already formed, or the ZrO_2_ are added as additional particles. Ionescu et al. [55] report that the porosity of the obtained materials increases with the addition of Zr. On contrary, when the Zr source is an active substance such as the Zr alkoxide, the porosity dramatically decreases because of the crosslinking effect of the heteroatom [55]. Acetylacetonate, however, is a complexing agent that hinders the formation of the mixed oxycarbide structure with a preference for an early formation of amorphous ZrO_2_ nanoclusters [56]. Some authors also report that the maintenance of the mesoporous structure even at pyrolysis temperatures of 1000 °C can be attributed to the decomposition of the Zr(Acac) used as a precursor, which was unable to form a mixed network with the polysiloxane [57]. The maintenance of the microporosity at high temperature in this work is attributed to the already observed phase separation, occurring together with the decrease of the Si–O–C bonds. The formation of a highly interconnected tridimensional network between the SiO_2_ and ZrO_2_ nanoclusters might also be responsible for the delay in the appearance of the transient micro-mesoporosity and the observed constant mesopore volume at temperatures beyond 700 °C [58]. The minimum content of Si–O–C bonds also prevents pore collapse and thus the appearance of micropores even at high pyrolysis temperature. In addition, this highly interconnected microporous network almost disappears after the HF etching (Table 3), suggesting that the ZrO_2_ nanoclusters could be somehow lixiviated in the digestion process because of the strong binding to the SiO_2_ domains. 

In addition, some authors have reported that the presence of the ZrO_2_ units inhibits the graphitization degree of the free carbon in SiZrOC materials [59]. In our materials, despite the decreased I_D_/I_G_ ratio in the pyrolyzed samples, after etching, this ratio remains constant at all the pyrolysis temperatures. Similarly, the combustion enthalpy of the carbon phase does not alter with the temperature, suggesting that there are no further changes in the free-carbon phase configuration. Liu et al. [59] suggest that the thermal stability of the Si–C bond is decreased in the presence of Zr, and therefore, the cleavage of these bonds during the redistribution reaction is facilitated. 

One of the most remarkable characteristics of the spectra of the Co-containing materials is the large intensity of the band assigned to C=C bonds (Figure 2d). Although it is still noticeable, this effect is not so evident when both the Zr and Co are introduced into the preceramic matrix. In the HF-etched samples, the same behavior is repeated, with a significant increase of the SiO_2_ interconnected in rings or chains (Figure 3b). In both cases, either when the Co is introduced alone or together with Zr, the I_D_/I_G_ ratio increases with the temperature either before or after etching. Here, we should consider the origin of the D band, which is attributed to the breathing mode of sp^2^ rings located at the edge of the graphite planes [60] and, as mentioned before, it becomes narrower due to the wet-etching process. 

It is also known that Co forms a solid solution with Si, resulting in nucleation sites for the SiC crystallization at high temperatures [61]. As mentioned above, contrary to Zr, which preferably forms ZrO_2_ nanoclusters, large Co crystallites appearing in Co-containing SiOC direct the microstructural characteristics of these samples [20,61]. The XRD patterns showing the formation of these crystals are shown in Appendix A. The multiple decomposition steps appearing in the DSC thermograms are attributed to crystalline cobalt carbides, which are not etched away [44]. In the presence of Zr, the temperature at which this decomposition takes place is slightly higher than in the case of Co alone and, as we observe in the Appendix A, the multiple carbidic forms are well revealed, suggesting that the ZrO_2_ nanoclusters that are formed from the decomposition of the Zr(acac) act as a protective phase for the stabilization of the unstable carbides.

## 5. Conclusions

HF etching results in a quite convenient strategy to study the structural changes occurring during the pyrolytic conversion of a polymeric or hybrid material subjected to any molecular modification. Nevertheless, it should be kept in mind which phases are more susceptible to the chemical etching and the structural changes induced by this treatment. The systematic study of these changes in a series of metal-modified polymer-derived materials allowed us to find that the highly interconnected tridimensional units in the SiO_2_ phase are the most susceptible to wet etching in such a way that the materials that are more favorable to produce this sort of arrangements will also be more attacked by fluorine. In a similar fashion, the carbon atoms located near these structures will also be etched away, thus provoking some changes in the remainder carbon phase.

By introducing Al atoms into the structure, AlOC bonds are formed even at low temperatures, thus increasing the crosslinking degree of the preceramic network. The I_D_/I_G_ ratio after etching increases with the pyrolysis temperature, whereas if no further treatment is performed, the trend of this ratio is exactly the opposite, suggesting that etching preferably affects the sp^3^ carbon. The formation of the tridimensional network formed by substitution of some Si atoms by Al, together with the presence of one type of carbon, might be responsible for the piezoelectric properties reported for this material. In the case of Zr incorporation, the relative amount of SiC units decreases with the Zr content because of the occurrence of phase separation during pyrolysis and the carbon phase almost shows no variation in its graphitization degree. The elimination of the SiO_2_ phase after HF etching occurs preferentially in these SiO_2_ clusters, arranged in a tightly interconnected tridimensional network, increasing the well-known thermal stability of the SiZrOC materials. Contrary to what occurs in the remainder metal-modified materials, the combustion enthalpy of the carbon phase remains constant at all the temperatures because of this increased thermal stability. In view of the Raman spectra, this carbon phase seems to be the more amorphous among all the prepared materials. The complexation effect of the acetylacetonate is also noticeable here since a delay in the formation of the transient mesoporosity is observed as well as the maintenance of the porous structure at the highest pyrolysis temperature.

Another area that has been investigated is the effect of incorporating a transition metal capable of forming a solid solution with Si together with Zr. In this case, the formation of small Co nanocrystals and the enhancement of the C=C bonds are the main characteristics of these materials. The oxidant decomposition of the carbon phase occurs in different stages depending on the pyrolysis temperature, indicating the transition of the carbon phase from low to high graphitization. Similarly, the trend of the I_D_/I_G_ ratio with the temperature before and after etching confirms the higher graphitization degree induced by the cobalt despite the presence of the Zr and its amorphization effect on the free-carbon phase.

## Figures and Tables

**Figure 1 materials-14-03276-f001:**
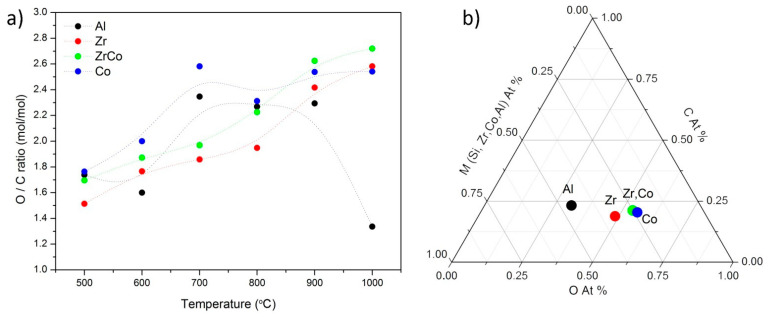
(**a**) O/C ratio of the samples and its evolution with pyrolysis temperature (dotted lines are drawn to guide eye) and (**b**) ternary diagram of the samples pyrolyzed at 1000 °C.

**Figure 2 materials-14-03276-f002:**
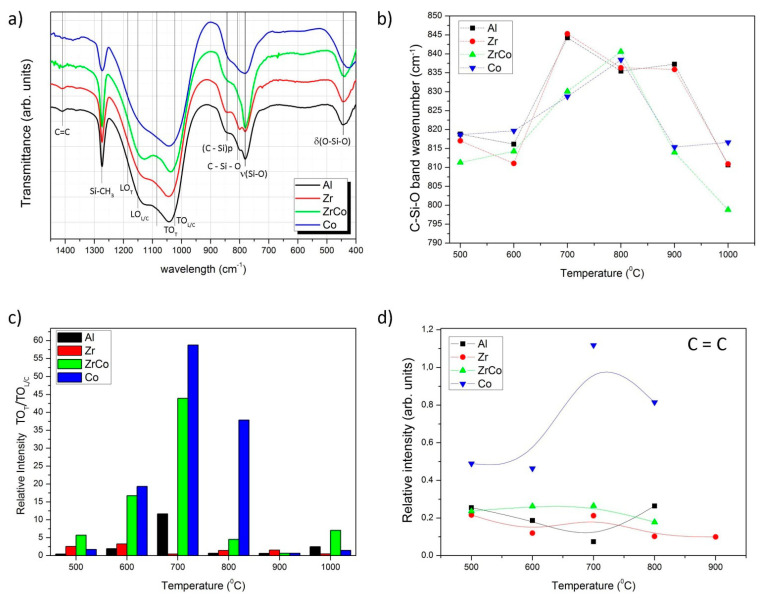
(**a**) FTIR spectra of the samples pyrolyzed at 700 °C; (**b**) position of the band corresponding to the formation of SiOC bonds as a function of the temperature; (**c**) relative intensity ratio of the bands assigned to the stretching mode of the superstructural rings forming tridimensional (T) and chains (L/C); (**d**) relative intensity of the bands assigned to the asymmetric stretching of double-bonded carbon. This intensity was normalized to the area of the band at 1080 cm^−1^.

**Figure 3 materials-14-03276-f003:**
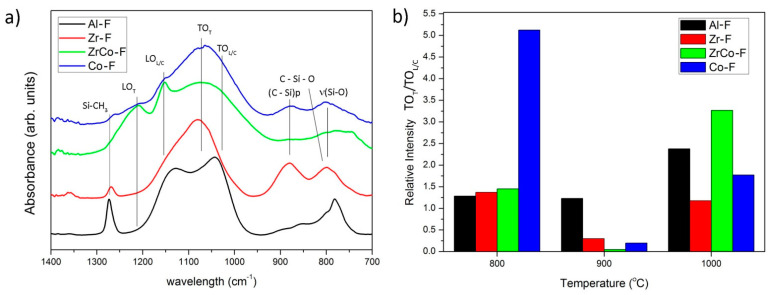
(**a**) FTIR spectra (in the absorbance mode) of the HF samples pyrolyzed at 900 °C and (**b**) the relative intensity ratio of the TO_4_/TO_6_ modes of the HF-etched samples.

**Figure 4 materials-14-03276-f004:**
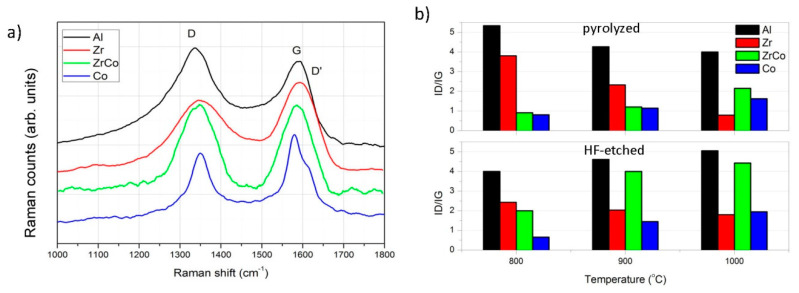
(**a**) Raman spectra of the samples pyrolyzed at 900 °C; (**b**) relative intensity of the D and G bands (ID/IG) of the Raman spectra of the pyrolyzed and HF-etched materials.

**Figure 5 materials-14-03276-f005:**
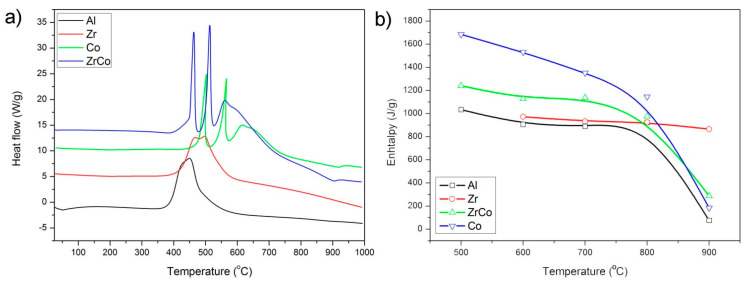
(**a**) Differential scanning calorimetry curves of the materials pyrolyzed at 600 °C; (**b**) oxidation enthalpy.

**Figure 6 materials-14-03276-f006:**
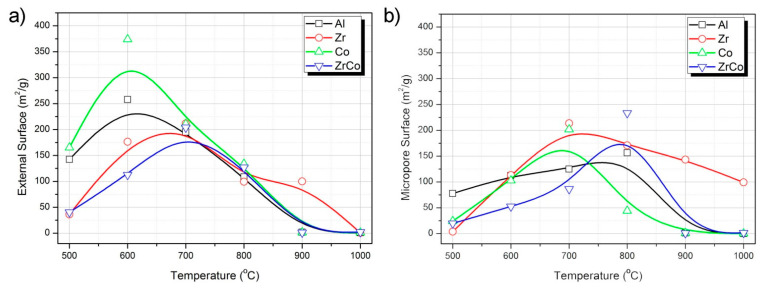
Surface area of the materials corresponding to (**a**) mesopores and (**b**) micropores.

**Table 1 materials-14-03276-t001:** Chemical composition of the studied materials (deviation ±0.05).

T (°C)	Al	Zr	Co	Zr/Co
500	SiAl_0.05_O_1.68_C_0.97_H_3.13_	SiZr_0.02_O_1.42_C_0.94_H_2.82_	SiCo_0.06_O_1.57_C_0.89_H_2.79_	SiZr_0.02_Co_0.03_O_1.59_C_0.94_H_3.02_
600	SiAl_0.05_O_1.18_C_0.74_H_2.31_	SiZr_0.02_O_1.52_C_0.86_H_2.41_	SiCo_0.07_O_2.18_C_1.09_H_3.75_	SiZr_0.02_Co_0.03_O_1.93_C_1.03_H_2.80_
700	SiAl_0.05_O_1.39_C_0.59_H_1.08_	SiZr_0.02_O_1.08_C_0.58_H_1.69_	SiCo_0.07_O_2.53_C_0.98_H_2.19_	SiZr_0.02_Co_0.03_O_1.99_C_1.01_H_2.36_
800	SiAl_0.05_O_1.55_C_0.68_H_0.70_	SiZr_0.02_O_1.43_C_0.73_H_1.12_	SiCo_0.06_O_1.85_C_0.80_H_1.83_	SiZr_0.02_Co_0.03_O_1.85_C_0.83_H_1.32_
900	SiAl_0.05_O_1.41_C_0.61_H_0.23_	SiZr_0.02_O_1.58_C_0.65_H_1.01_	SiCo_0.06_O_1.98_C_0.78_H_0.11_	SiZr_0.02_Co_0.03_O_1.94_C_0.74_H_1.18_
1000	SiAl_0.05_O_0.68_C_0.51_H_0.04_	SiZr_0.02_O_1.50_C_0.58_H_0.03_	SiCo_0.07_O_2.16_C_0.85_H_0.07_	SiZr_0.02_Co_0.03_O_2.35_C_0.86_H_0.04_

**Table 2 materials-14-03276-t002:** Temperatures (°C) of the decomposition of the materials when heat treated in air atmosphere (n/d stands for none determined).

T (°C)	Al			Zr			Co				Zr/Co		
	Pyr	HF Etch		Pyr		HF Etch	Pyr		HF Etch		Pyr		HF Etch
600	450	n/d		465	495	n/d	465	515	n/d		500	565	n/d
700	495	n/d		500		n/d	470	600	555		525	575	n/d
800	545	n/d		550		n/d	550	585	580		545	600	535
900	650	355	550	550		525	475		520	575	480		545

**Table 3 materials-14-03276-t003:** Specific surface area of the studied materials as obtained from the BET method applied to the nitrogen adsorption isotherms (the asterisk in Co * refers to already published data reprinted from [44] with permission from Elsevier).

SSA m^2^/g				
T (°C)	Al	Zr	Co *	Zr/Co
500	220	40	190	60
600	370	390	475	165
700	320	325	335	360
800	265	270	255	290
900	3.0	243	2.3	2.0
1000	0.5	100	0.5	3.5
SSA m^2^/g after HF etching				
700	--	1.2	2.0	--
800	--	41	43	--
900	91	3.3	74	17

## Data Availability

The data presented in this study are openly available.

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
