# Peer review of "Insights into the Microstructural Evolution Occurring during Pyrolysis of Metal-Modified Ceramers Studied through Selective SiO2 Removal"

_materials, 2021, doi:10.3390/ma14123276_

Round 1

Reviewer 1 Report

comment on materials-1244516

This is an interesting work regarding how to achieve the polymer pyrolysis derived ceramic composites from ceramers. Some comments are attached as follows.

  1. Abstract, line 5: “From the structural an textural characterization, it was…”, “an” could be a misspelling for “and”.
  2. Conclusion, final line: “despite the presence of the Zr and its amorphization effect on the free carbon phase”, a period is missing after “phase”.
  3. The keyword “mesoporosity” appears twice in abstract and conclusion. This indicates that mesoporosity is an important phenomenon. In addition to specific surface area data shown in table 3, if possible, I suggest authors provide some SEM micrographs as the evidence.

Author Response

Reviewer #1: Abstract, line 5: “From the structural an textural characterization, it was…”, “an” could be a misspelling for “and”.

Authors: The authors have performed a thoughtful revision of the overall manuscript. And some mistakes as the one that reviewer noticed were corrected. Thanks for the observation

Reviewer #1: Conclusion, final line: “despite the presence of the Zr and its amorphization effect on the free carbon phase”, a period is missing after “phase”.

Authors: This has been corrected

Reviewer #1: The keyword “mesoporosity” appears twice in abstract and conclusion. This indicates that mesoporosity is an important phenomenon. In addition to specific surface area data shown in table 3, if possible, I suggest authors provide some SEM micrographs as the evidence.

Authors: The authors are in complete agreement with the reviewer in the point that the mesoporosity is an important phenomenon occurring in this kind of materials. We have included in the supplementary information, the pore size distributions of the HF etched materials thus demonstrating that still after the chemical attack, some mesoporosity still remain in the materials. We have included this info instead of the suggestion of including the SEM micrographs because the mesoporosity is too small to be directly observable by SEM. We have suggested a more appropriate technique to show this mesoporosity.

Reviewer 2 Report

The authors present in their manuscript a study related to the evolution of metal-modified silicon oxycarbides and an ambitious attempt to offer a universal description of the temperature-dependent evolution of metal-modified polysiloxane-based precursors upon ceramization. The topic is indeed highly interesting and relevant for the scope of the Journal. However, I think that the authors missed their ambitious goal to offer a description for the evolution of the metal-modified precursors which can hold for any metal used for their modification, as I will try to elaborate on in the following:

  1. Firstly, the rationale for the selection of the metal considered in the study (Al, Zr, Co) is not obvious. The authors are encouraged to elaborate on this aspect in a more comprehensive and crisp way. Also, a minor concern relates to the fact that the introduction of the metals within the molecular structure of the polysiloxane occurs by using either metal acetyl acetonate or acetates. While both ligand acac and ac possess many similarities, their thermal decomposition (and consequently the thermal decomposition of the metal-precursor-modified polysiloxanes) may differ and thus a direct comparison of the metal-modified precursors is not straight forward. For instance, the discussion related to the effect of the metal on the O/C ratio is then highly questionable if metal-acetylacetonate-modified polysiloxanes are compared to metal-acetate-based precursors.
  2. Concerning the discussion of the evolution in the Al-modified precursor, it is not clear how the authors expect the Si-O-C linkages to convert/rearrange into Si-O-Si / Si-O-Al – this is not substantiated by any experimental data or literature quotation and thus just highly speculative. Also other statements of the authors concerning the Si-Al-O-C system are doubtful or at least highly speculative. For instance, the authors state that the free carbon phase in the Si-Al-O-C system is strongly affected by the Al incorporation into the SiOC glassy network – this is in my opinion not clear / obvious and may not be necessarily correct – also, the authors do not give any experimental support for their statement.
  3. Furthermore, the authors are encouraged to perform their comparative discussion by considering the ternary system (metal-free) Si-O-C. The reference SiOC sample series is missing in the present study mand thus many of the conclusions made by the authors are not obvious or even questionable.
  4. Also, I miss X-ray diffraction data of the investigated samples. The authors repeatedly consider the appearance of crystalline phases in their samples (e.g., Co, Co carbide(s), Al (oxy)carbides, ZrO2 etc) – however, there is no XRD pattern shown in the manuscript (this holds also for the Supplementary Information). Additionally, TEM may help too in order to rationalize the appearance and evolution of crystalline phases in the studied systems, this was also not considered in the present study.
  5. Various aspects discussed in the manuscript requires careful reassessment. For instance, the authors state in their work that the Al incorporation increases the cross-linking in the SiOC glassy network (– there is no experimental proof given by the authors to support this) – I doubt that this is correct; typically, Al is being indeed considered as network former e.g. in aluminosilicate glasses; however, I would like to stress two critical point here: (i) firstly, it is not clear (and not shown by the authors) how much Al is incorporated within the SiOC glassy network and how much if it is present as secondary phases - here, NMR data would help indeed to elucidate this question; and (ii) Al prefers the coordination number 4 as glass former, with small fractions of penta- and hexacoordinated Al in the glassy structure – this, in addition to the fact that the Al : Si ratio in the system prepared by the authors is really low, makes the statement of the authors highly doubtful.
  6. As an additional criticism (minor): the authors claim in their Introduction that the free carbon phase in the Si-Al-O-C emerges as a result of VLS mechanism – this is for me rather unusual statement and highly questionable.
  7. Also, the discussion of the authors putting the Al incorporation into the Si-O-C glass in direct correlation to the piezo-dielectric behavior found in SiAlOC is highly doubtful. Most probably, the authors had the intention to correlate the Al incorporation and its assumed effect on the segregated carbon phase in SiAlOC (which is not clear, neither in the present manuscript, nor in the cited work) with the mentioned piezo-dielectricity on SiAlOC – this is in my opinion not sensible.

Author Response

Reviewer 2: Firstly, the rationale for the selection of the metal considered in the study (Al, Zr, Co) is not obvious. The authors are encouraged to elaborate on this aspect in a more comprehensive and crisp way.

Authors: We have restructured the introduction part in an attempt to better explain the selection of the substituting heteroatoms. On the one hand, Al and Zr substituting SiOC are some of the most studied heteroatoms (not considering SiBOC ceramers, which, from the point of view of the Author, form a complete distinct family of polymer derived ceramics which should be considered separately). Al forms mullite and Zr forms ZrO2 thus reducing the relative proportion of Si-O bonds (or increased SiC). Co, on its side, affects more to the formation of the C nanodomains because of the formation of transient cobalt carbides (See ref 62)

Reviewer 2: Also, a minor concern relates to the fact that the introduction of the metals within the molecular structure of the polysiloxane occurs by using either metal acetyl acetonate or acetates. While both ligand acac and ac possess many similarities, their thermal decomposition (and consequently the thermal decomposition of the metal-precursor-modified polysiloxanes) may differ and thus a direct comparison of the metal-modified precursors is not straight forward. For instance, the discussion related to the effect of the metal on the O/C ratio is then highly questionable if metal-acetylacetonate-modified polysiloxanes are compared to metal-acetate-based precursors.

Authors: The authors completely agree with the reviewer in the sense that using different precursors may lead to different O/C ratios because of the different thermal decomposition. In fact, in the discussion part, it is referenced the work of Ionescu et al (ref 55) where they claimed the absence of the formation of a mixed network in Zr(acac) containing polymer derived ceramics due to the thermal decomposition of the organic ligand. In this work, Acac has been used as a common ligand in both Al and Zr incorporation (which are known to affect the number of the Si-O bonds) while in the case of Co, which primarily affect to the C phase, Ac ligand was used. In the revised version of the manuscript, we have highlighted this point, by including a sentence where we recognize that this O/C ratio might be different when using different ligands.

Reviewer 2: Concerning the discussion of the evolution in the Al-modified precursor, it is not clear how the authors expect the Si-O-C linkages to convert/rearrange into Si-O-Si / Si-O-Al – this is not substantiated by any experimental data or literature quotation and thus just highly speculative. Also other statements of the authors concerning the Si-Al-O-C system are doubtful or at least highly speculative. For instance, the authors state that the free carbon phase in the Si-Al-O-C system is strongly affected by the Al incorporation into the SiOC glassy network – this is in my opinion not clear / obvious and may not be necessarily correct – also, the authors do not give any experimental support for their statement.

Authors: The formation of the Si-O-Al network has been already described by several authors in PCS derived materials. The most plausible mechanism is via dehydrogenation of Si-H bonds or at the expenses of the Si-CH3 residues (probably with the intermediate formation of Si-H). In the revised version of the manuscript we have included the reference 48 by Zhang et al who studied, by NMR the amounts Si−O−Al and Si−O−Si groups in PACS. With regard to the carbon phase modification, in Figure 2d, it is shown that the formation of C=C bonds is enlarged. Nevertheless, to support this finding, we have also included some other references of authors which also found a modification of the free carbon phase because of the inclusion of the A heteroatom within the preceramic network and attributed to the cationic substitution of the Al atoms in the glass network (ref 47).

Reviewer 2: Furthermore, the authors are encouraged to perform their comparative discussion by considering the ternary system (metal-free) Si-O-C. The reference SiOC sample series is missing in the present study and thus many of the conclusions made by the authors are not obvious or even questionable.

Authors: The reviewer is right in affirming that we did not compare our results with the metal-free material. The pyrolysis of the MK polymer has been described since long in many different articles (e.g. DOI:10.1016/j.jeurceramsoc.2003.10.016). This article is focused on the effect on each heteroatom on the evolution of the microstructure of SiMOC ceramers rather than a comparison with the metal-free material.

Reviewer 2: Also, I miss X-ray diffraction data of the investigated samples. The authors repeatedly consider the appearance of crystalline phases in their samples (e.g., Co, Co carbide(s), Al (oxy)carbides, ZrO2 etc) – however, there is no XRD pattern shown in the manuscript (this holds also for the Supplementary Information). Additionally, TEM may help too in order to rationalize the appearance and evolution of crystalline phases in the studied systems, this was also not considered in the present study.

Authors: According to reviewer´s suggestion, we have included the XRD analysis of the HF materials in the supplementary info (in the case of the non-etched samples, the XRD were typical of amorphous glasses).

Reviewer 2: Various aspects discussed in the manuscript requires careful reassessment. For instance, the authors state in their work that the Al incorporation increases the cross-linking in the SiOC glassy network (– there is no experimental proof given by the authors to support this) – I doubt that this is correct; typically, Al is being indeed considered as network former e.g. in aluminosilicate glasses; however, I would like to stress two critical point here:

(i) firstly, it is not clear (and not shown by the authors) how much Al is incorporated within the SiOC glassy network and how much if it is present as secondary phases - here, NMR data would help indeed to elucidate this question;

(ii) Al prefers the coordination number 4 as glass former, with small fractions of penta- and hexacoordinated Al in the glassy structure – this, in addition to the fact that the Al : Si ratio in the system prepared by the authors is really low, makes the statement of the authors highly doubtful.

Authors: The author is right in affirming that we did not show how much A was incorporated into the preceramic network. In fact, we recognize that the chemical composition was calculated by assuming that the ratio Si/M (M=Al, Zr, Co) is maintained constant during the pyrolysis and this proportion does not vary with respect of the precursor materials (see Table 1 in the manuscript).

With regard to the first point, when considering a highly crosslinked network we are not referring exclusively to the Si-O-Al linkages but the silica network as well. We have calculated the amount of highly crosslinked SiO2 superstructural units  (appearing in the FTIR at 1030 cm-1) and the tridimensional SiO2 network (appearing at about 1080 cm-1) from the FTIR spectrum and from this, we led to the conclusion that the incorporation of each heteroatom produces a different amount of highly crosslinked SiO2 structures or loosely bonded SiO2. We cannot quantify the amount of these units, in the same since we have not quantify the amount of Al incorporated into the silica network (as a glass former). What is clear is that the incorporation of Al or Zr leads to a different amount of superstructural SiO2 than when the Co is included as the heteroatom (see Figure 2c). As the reviewer noticed, the NMR would help to the dilucidation of these structures. In future works, we will consider this technique to provide a semiquantitative amount of heterolinkages.

The second point relates with the coordination of the Al within the glass network. The reviewer highlights that the Al might adopt several coordination environments in silicate glasses depending on its concentration. Indeed, in our materials, the amount of Al is quite low, so it is expected a 4 coordination. We haven´t proved this because we are more focused on the possibility to etch the silica phase. As it was stated in the manuscript, “…In the HF-etched samples, the TOT to TOL/C ratio decreases with respect to their corresponding non-etched counterparts indicating that, the SiO2 units which were preferentially etched are the ones in interconnected rings or chains…”. This indicates that in the case that Al is incorporated in the SiO2 network, it will be preferably linked to the tridimensional SiO2 as a network former (i.e. in the 4-coordination).

Reviewer 2: As an additional criticism (minor): the authors claim in their Introduction that the free carbon phase in the Si-Al-O-C emerges as a result of VLS mechanism – this is for me rather unusual statement and highly questionable.

Authors: The authors performed a revision of the existing literature when preparing this manuscript and, unfortunately we didn´t find any work which contradicts the work performed by Yu et al (cited reference nº 14). We want to highlight here that we are not referring to the free carbon phase but carbon exclusively (which can be in form of fibers, nanowires, or any other morphology). The VLS mechanism is not an usual mechanism for the segregation of the free C phase, of course, but it can be on the formation of different carbon nanostructures.

Reviewer 2: Also, the discussion of the authors putting the Al incorporation into the Si-O-C glass in direct correlation to the piezo-dielectric behavior found in SiAlOC is highly doubtful. Most probably, the authors had the intention to correlate the Al incorporation and its assumed effect on the segregated carbon phase in SiAlOC (which is not clear, neither in the present manuscript, nor in the cited work) with the mentioned piezo-dielectricity on SiAlOC – this is in my opinion not sensible.

Authors: It wasn’t of the intention of the authors to correlate the segregation of the carbon with the piezoelectric behavior and we recognize that probably the sentence was not expressed in the right manner. In our discussion, we have introduced the origin of the piezoelectricity of this kind of materials through the following sentence:

“This piezo-dielectricity of the material results from the formation of space charges along the interface between the relatively insulating ceramic phase and the more conductive free-carbon phase [15].”

Reviewer 3 Report

Please see the attached comments.

Author Response

Reviewer 3: Please address the following formatting issues:

(a) Although the manuscript is very well-written, it contains multiple grammatical and spelling errors and requires editing/proof reading.

(b) Ensure all acronyms, abbreviations, and chemical formulae are defined when they first appear in the abstract or main text.

(c) Correct form of chemical formulae should be used (e.g., SiO2, and ZrO2 are not acceptable). Also, avoid switching back and forth between using chemical formulae and spelling the name of chemicals.

(d) Try to be consistent in your terminology as much as possible. For example, avoid switching back and forth between “heteroatom” vs. “heterogeneous element” OR “sample” vs. “system” vs. “material” OR “ZrCo” vs. “Zr,Co” vs. “Zr+Co” vs. “Zr/Co”.

(e) Remove the underline from all degree symbols.

(f) It is recommended to use “arb. units” instead of “a.u.” because the latter stands for atomic units.

(g) In figures, try to use consistent symbols for different samples to avoid confusion.

Authors: We have performed a complete revision of the manuscript by correcting the spelling issues and grammar mistakes as well as the formatting issues addressed by the reviewer. Thanks for noticing these issues which were unadvertised form our side

Reviewer 3: In the abstract, please clarify that the samples contain 5 wt.% of heteroatom to Si.

Authors: In the abstract, we have clarified that the amount of heteroatom was included with respect to the total amount of Si

Reviewer 3: Line 147 states “it have been estimated that the ratio Si/M (M=Al, Zr, Co) is maintained constant during the pyrolysis and this proportion does not vary with respect of the precursor materials”. However, based on the values reported in Table 1, molar fraction of Co/Si in the Co sample appears to change with temperature. Please clarify.

Authors: In the calculation of the chemical formulae, we have clarified that we have considered this 5% heteroatom constant. No etching, no removal of heteroatom has been considered thus the total amount (in weight) is maintained during the pyrolysis.

Reviewer 3: With respect to Fig. 1:

  1. It is stated that “in the case of Al, there is a maximum in the O/C ratio at the intermediate pyrolysis temperatures (600 – 800 ºC)”. However, maximum values for the Al sample are observed at 700 – 900 ºC.

Authors: This has been corrected

  1. It is stated that “when the heterogeneous element is a transition metal, it is observed a gradual increase of the O/C ratio with the pyrolysis temperature”. However, the O/C ration for the Co shows a maximum at 700 ºC.

Authors: Actually, we are considering a trend rather than individual points. In this case, we can assume that this maximum falls within the experimental error and the trend is a gradual increase of the O/C ratio with the pyrolysis temperature

  1. It is stated that “ratio is also slightly higher when the Co is incorporated into the preceramic network”. However, this statement is only true at temperatures ≤800 ºC.

Authors: Here,  we have clarified that the incorporation of Co may be alone or in combination with Zr

  1. The discussion of Fig. 1(b) does not provide any additional insights compared to Fig. 1(a)? The authors should either expand the discussion of this figure or remover it from the manuscript.

Authors: We have clarified that Figure 1 reinforces our previous observation that the incorporation of Co increases the O/C ratio

Reviewer 3: With respect to Fig. 2:

  1. It is stated that “In the low-frequency region, the d(O-Si-O) centered at 470 cm-1 is shifted to lower wavenumbers in the sample containing Co indicating a highly stressed structure”. First, please describe the factors contributing to this stressed structure. Second, similar observations can be made for nearly all the samples as the position of O-Si-O decreases at lower pyrolysis temperature. Is this also an indication of more stressed state at lower temperatures? Please explain.

Authors: We have substituted the term “stressed” to “tensioned”, which might be more clear to the readers. We have also clarified that a stressed structure is the one containing distorted O-Si-O angles with respect to the ones normally found in silica.

  1. 2(b), please rename the y-axis to “C−Si−O Band Wavenumber (cm-1 )”.

               Authors: According to the reviewer suggestion, this has been modified

  1. It is stated that “Fig. 2 b presents the relative position of this band calculated by means of a deconvolution analysis where we can appreciate a dramatic increase in the samples containing Co and Zr+Co”. Does this imply that the increase in the Co and ZrCo samples is more drastic than the Al or Zr samples? Page 2

Authors: This sentence was wrongly placed there. The authors acknowledge the reviewer for noticing that. This sentence was referred to the Figure 2c where a dramatic increase in the intensity of the TO(T)/TO(LC) band ratio was observed.

  1. It is stated that “If the incorporated heteroatom is Co or Zr+Co, it is produced a gradual shift to high wavenumbers when the pyrolysis is carried out up to 800 ºC and then drops to the normal values”. Please clarify what is considered as a normal value.

Authors: The authors have clarified that the “normal” value is 800-810 cm-1 for SiO2 and SiOC

  1. In Fig. 2(d), how did you normalize the intensity of the C=C peaks?

Authors: In the figure caption we have included that the intensity of this band (area) was normalized to the band centered at 1080 cm-1

  1. It is stated that “it is possible to distinguish the multiple combinations of the Si-O rings and Si-OX”. Please clarify what “X” is.

Authors: The authors have indicated that X can be substituted by the heteroatom symbol

  1. It is stated that “in Fig. 2 c and it is observed that the relative intensity of this ratio remains mostly constant when either Al or Zr were incorporated into the hybrid network”. However, the relative intensity of the Al sample shows a noticeable increase at 700 ºC. Please clarify.

Authors: It is true that this band seems to increase its intensity with respect to the previous and following pyrolysis temperatures but in reality it increases progressively up to 700 ºC and then decreases. This increase is not so noticeable than in the case of the Co-containing materials. This has been also included in the text in the result part and in the discussion section:

Moreover, the condensation mechanism of the Al-containing samples occur via dehydrogenation reactions [45,46] with the formation of the C=C bonds at 700 °C (Fig. 2 d) [47]. At this temperature also, the relative intensity of the TOT/TOLC band is the highest. Zhang et at. [48] reported the formation of Si-O-Al bridges during the pyrolysis at the expenses of the Si-H bonds. At this temperature, and accompanied with the decrease of the Si-H band in the FTIR spectra (2100 cm-1), the position of the Si-O-C band is shifted from 817 cm-1 to its maximum value, at 845 cm-1 which is assigned to the symmetric stretching vibration of the Si–O–Si and Si–O–Al linkages [49].

Reviewer 3: Figs. 2 and 3:

  1. As indicated by the authors, the FTIR peak around 1200 cm-1 corresponds to the longitudinal optical (LO) modes. This peak is reported to be an indication of porosity in silica-based materials and thus the ratio of LO/TO intensity could be used as an indication of porosity. Also, the shift in the TO spectral center is reported to be related to the Si−O−Si bond angle and therefore compaction of the silica network [Ref: S.A. Shojaee et al., Journal of Materials Science 52, no. 20 (2017): 12109- 12120]. Please comment whether these arguments are true for the current study.

Authors: The reviewer is absolutely right in the assertion that this band can be related to the porous structure of the silica. This correlation was reported in 2003 by P. Innocenzi that stated that the pores induce a deformation in the Si–O–Si bonds that gives rise to longer bridging angles and weakly longer Si–O bonds compared to bulk glassy silica. Especially in the Co-containing materials, a modification of the O-Si-O bridging angle was observed and this was not necessarily correlated with the porosity (see Table 3). In this case, we consider that the correlation of the TO/LO bands with the porosity does not apply and it is more important the effect of the heteroatoms and the structure of the SiOC network than the deviation of the Si-O-Si angles because of the presence of the pores.

  1. Line 223 states “The most dramatic change in the TO T/TO L/C ratio is found in the samples containing Co where the relative proportion of tridimensional units decreases in a continuous manner with the temperature”. First, do you mean continuous or consistent? Second, it is unclear whether this statement refers to the differences between the as-pyrolyzed and etched states or differences between the Co samples pyrolyzed at different temperatures.

Authors: According to the reviewer suggestion, we have explained in the text that this decrease occurred in the HF-etched samples. Accordingly, we have omitted the expression “continuous manner” and now the text indicates that the relative proportion of tridimensional units decreases with the temperature.

Reviewer 3: Fig. 4(a) shows the spectra of the etched samples. Please edit the figure legend to correctly label the samples. Also, please explain if background subtraction was performed prior to finding the I(D)/I(G) ratio.

Authors: The legend has been edited. In addition, we have included in the text that the background subtraction was performed after the band deconvolution.

Reviewer 3: Line 266 states “The samples containing Zr as the single heteroatom decompose in an unique step”. Do you mean unique or single step?

Authors: Thanks for the comment. It was obviously a “false friend” form our native language. We meant single step.

Reviewer 3: In Fig. 5(b), why is there no data point for the Al sample pyrolyzed at 800 ºC?

Authors: A closer look at the figure, the reader can be appreciated that this data point is surprisingly overlapped by the Zr point.

Reviewer 3: With respect to Table 2:

(a) This table throws a lot of information at the reader without describing the chemical reactions that correspond to each peak position.

Authors: The reviewer is right in this affirmation. In all the cases, however, they are related to the oxidation of the material (notice that the DTA was carried out in oxidant atmosphere). We didn´t include the reactions since they should be better studied by other complementary techniques such as TGA/MS

(b) For the Co sample, it is difficult to distinguish between the values reported for the as-pyrolyzed and etched states.

               Authors: A dotted line has been included in the table to facilitate the observation

(c) What does n/d stand for?

Authors: We have added to the legend in the table that n/d stands for non determined

(d) Please double check the peak positions reported for the as-pyrolyzed Zr samples at 700 and 800 ºC.

Authors: Double checking the peak positions, we have noticed an error in the color assignations of in curves in Figure S4. Now, it has been corrected. Thanks for that.

(e) At some temperatures, more than two peaks are observed in the DSC curves of the Co and ZrCo samples but only two peak positions are reported in the table.

Authors: We have included some more information with regard to the peaks appearing in the DSC of the materials containing Co. In the text we have specified that although there are multiple peaks because of the different forms of carbon only the most prominent peaks are referred.

Reviewer 3: What does the asterisk refer to in the header of Table 3?

Authors: Many thanks to the reviewer for noticing we didn´t include the menaning of the asterisk. With this asterisk, as mentioned in the revised version of the manuscript, we refer to already published data (reference has been also included)

Reviewer 3: Line 302 states “With regard to the micropore surface area (Fig. 6 b), … , it can be observed that the amount of micropores gradually increases to reach the maximum in the materials containing Al”. Are you implying that the overall maximum microporosity occurs in the Al sample?

Authors: We have modified this sentence and now it is stated that the amount of micropores gradually increases until reaching its maximum at 700 ºC.

Reviewer 3: Page 3. Line 329 states “… thus decreasing the free carbon amount and the observed decrease of the O/C ratio at temperatures beyond 800 ºC”. However, according to Fig. 1(a) the O/C ratio appears to be similar at 800 and 900 ºC.

               Authors: Exactly. At 800 and 900 ºC, it is similar.

Reviewer 3: With respect to supplementary material:

 (a) Please label important peaks in the reported FTIR and Raman spectra.

               Authors: The most prominent bands in the FTIR and Raman spectra have been labeled

(b) In Fig. S2, please use consistent tick marks for the x-axis of the reported plots. Also, the x-axis of Co samples pyrolyzed at 500 – 1000 ºC is not fully visible.

               Authors: These issues have been corrected in the revised version of the manuscript

(c) If possible, include the DSC curves of the etched samples.

               Authors: This has been done and include

Round 2

Reviewer 2 Report

The authors performed appropriate revision of their manuscript which is recommended for publication.